# Understanding the Impact of Epilepsy and Depression on Sleep Disorder: Beyond Associations

Vasundhara Acharya*, Bülent Yener*, Madeline C Fields[†], Lara V Marcuse[†]

*Rensselaer Polytechnic Institute, Troy, NY, USA
[†]Icahn School of Medicine at Mount Sinai, Mount Sinai Hospital, NY, USA

*Abstract*—Epilepsy, sleep disturbance (SD), and major depressive disorder (MDD) often co-occur and are further complicated by antiseizure medications (ASMs). Association-based methods struggle to untangle their causal links. Using National Health and Nutrition Examination Survey (NHANES) data from 2015–2020, we encoded unordered categorical variables via a variational autoencoder–principal component analysis (VAE-PCA) pipeline, learned a consensus directed acyclic graph (DAG) through an adaptive-bootstrapped ensemble of causal structure discovery (CSD) algorithms, and estimated backdoor-adjusted effects with generalized linear models (GLMs) and refutation tests. We observed direct average treatment effects (ATEs) of epilepsy on SD (0.362), MDD on SD (0.167), and ASMs affecting sleep architecture on SD (0.206), but no significant links between ASMs and MDD or between epilepsy and MDD. VAE embeddings highlighted population heterogeneity. While providing valuable hypotheses, the cross-sectional design limits definitive causal claims. These findings are a foundation for future longitudinal studies incorporating objective measures and advanced methods.

*Index Terms*—Epilepsy; Sleep Disturbance; Causal Structure Discovery;Directed Acyclic Graph; Average Treatment Effect

## I. INTRODUCTION

Epilepsy, sleep disorders (SD), and major depressive disorder (MDD) represent significant, co-occurring global health concerns. In 2021, an estimated 51.7 million people had epilepsy globally [1], while depression is a leading cause of disability [2]. The complex, often bidirectional, relationships between these conditions are well-documented. The epilepsy-sleep connection can form a vicious cycle, where nocturnal seizures fragment sleep and disrupted sleep, in turn, lowers seizure thresholds [3], [4]. Similarly, depression is a frequent and under-recognized comorbidity in people with epilepsy (PWE), associated with diminished quality of life and potentially distinct neuropathological underpinnings [5], [6]. The depression-sleep link is also mutually exacerbating. For instance, insomnia is a demonstrated risk factor for MDD [7]. This complex triad is further complicated by anti-seizure medications (ASMs), which are essential for seizure management but can independently alter both sleep architecture and mood [4], [8]. While large-scale observational datasets like NHANES capture these co-occurrences, conventional statistical methods primarily quantify associations, struggling to disentangle true causal pathways from confounding, especially with cross-sectional data. Moving beyond association to infer potential causality is a critical research challenge. While Randomized Controlled Trials (RCTs) are the gold standard [9], their application is often limited by cost and practical difficulties in these complex systems. To address this, our work uses an ensemble framework for Causal Structure Discovery (CSD), a strategy known to improve the accuracy and robustness over single algorithms [10]–[12]. Our approach uses adaptive bootstrapping to generate a stable Directed Acyclic Graph (DAG) and introduces a Variational Autoencoder (VAE) pipeline to encode categorical variables.

The primary objective of this work is not to establish definitive causal proof but to generate robust, data-driven, and testable hypotheses regarding the complex causal interplay between epilepsy, SD, MDD, and the effects of ASMs. By identifying potential causal links from rigorous observational analysis, we aim to provide a structured foundation to guide future experimental or longitudinal investigations.

The main contributions of the work are as follows:
- We curated the NHANES dataset, including key health and medication indicators, and compressed its categorical covariates that lack inherent order using a VAE pipeline.
- We learned a consensus DAG via an adaptive bootstrapping ensemble of CSD methods.
- We estimated backdoor-adjusted potential causal effects using GLM and tested their robustness.

## II. METHODOLOGY

### A. Dataset Preparation

We used data from the NHANES 2015–March 2020 pre-pandemic cycles. Epilepsy was determined by cross-referencing prescription medication data with ICD-10-CM codes for epilepsy or seizure prevention (G40, G40.P). Sleep Disturbance (SD) was indicated by a score below 2 on a composite sleep score, computed as per [13]. Major Depressive Disorder (MDD) was operationalized as a score $\geq$ 10 ( threshold chosen to reflect significant depressive burden and functional impairment) on a custom 9-item index (summing DPQ010-DPQ050, DPQ070-DPQ090, and DPQ100 (as we prioritized measuring functional impairment) derived from the PHQ-9 screener. Covariates included demographics (age, race, education, marital status, poverty-income ratio) and clinical factors (smoking status, BMI, hypertension, diabetes, CHD, and asthma). A key feature of our model was the specific handling of antiseizure medications (ASMs). We derived three binary flags for each participant's ASM usage based on the drug's known pharmacological properties: used primarily for Epilepsy (E), Sleep-inducing effects (S), or mood-altering/Depressive side effects (D). The specific medications

and their classifications are detailed in Tables I and II. To prevent potential biases associated with simple data imputation methods (mean and median), records with missing values for key variables, namely BMI, CHD, Hypertension, and other covariates, were removed from the final analytical dataset, similar to [13]. This process resulted in a final analytical sample of 7,217 participants (unweighted count). To derive nationally representative estimates of the prevalence for epilepsy, depression, and sleep disorders within the U.S. population, we utilized survey weights from the NHANES. Specifically, we employed the combined Interview and MEC weights, as the assessment of these conditions relied on data collected during MEC examinations. To combine data, the respective full-sample MEC weights were adjusted by multiplying the 2015-2016 weights by 2/5.2 and the 2017-March 2020 weights by 3.2/5.2, as recommended in [14]. The weighted count for each condition was then calculated using equation 1.

$$WeightedCount = \sum_{i=0}^{n}(MEC_i \times Status_i) \qquad (1)$$

where $MEC_i$ is the adjusted combined MEC weight for individual $i$, and $Status_i$ is an indicator variable. The survey weights indicated approximately 819,747 individuals with epilepsy (130,623,651 without). For depression, an estimated 14,764,333 individuals were affected (with 116,679,064 unaffected), and for sleep disorders, 32,247,706 were estimated to be affected (while 99,195,691 were not). Detailed data statistics are available upon request.

TABLE I: Definitions of variables selected for analysis

| Variables | Description |
| --- | --- |
| SEQN | Responded Sequence Number |
| RIAGENDR | Male/Female |
| Age_Category_4G | Age Categories 20-40, 40-60, >60 |
| RIDRETH1 | Race |
| DMDEDUC | Education Level |
| DMDMARTL_3 | Marital Status |
| INDFMPIR_2G | Income-Poverty Ratio <2 or >=2 |
| BMI_Categories | BMI Categories based on WHO guidelines |
| SD | Patient has Sleep disturbance/Not |
| DIABETES_2G | Patient has Diabetes/Not |
| SMQ_2G | Patient is a Smoker/Not |
| ASTHMA_2G | Patient has Asthma/Not |
| CHD_2G | Patient has Coronary Heart Disease/Not |
| MDD | Patient has Major depressive disorder/Not |
| HYPERTENSION_2G | Patient has Hypertension/Not |
| ASM1 | Intake of ASMs (Binary indicator) |
| ASM2 | Intake of ASMs (alter sleep architecture (Binary indicator)) |
| ASM3 | Intake of ASMs (have mood-altering properties (Binary indicator)) |
| EPILEPSY_2G | Patient has Epilepsy/Not |

## B. Encoding Categorical Variables

High-cardinality nominal categorical covariates, such as detailed race/ethnicity groupings, pose challenges for standard encoding. One-hot encoding can create sparse, high-dimensional feature spaces [15] [16], while simple label encoding is unsuitable as it misleadingly imposes a false order where none exists. While target-based encoding is a powerful

TABLE II: Antiepileptic Drugs and Their Additional Properties

| Drugs | Flag | Drugs | Flag |
| --- | --- | --- | --- |
| CARBAMAZEPINE | D,E | FELBAMATE | E |
| VALPROIC ACID | D,E | GABAPENTIN | E,S |
| PHENYTOIN | E | LAMOTRIGINE | D,E |
| DIAZEPAM | D,E,S | DIVALPROEX SODIUM | D,E |
| LORAZEPAM | E,S | TOPIRAMATE | E |
| CLONAZEPAM | D,E,S | LEVETIRACETAM | E |
| ETHOSUXIMIDE | E | OXCARBAZEPINE | D,E |
| PHENOBARBITAL | E | ZONISAMIDE | E |
| PREGABALIN | E | LACOSAMIDE | E |
| RUFINAMIDE | E | ESLICARBAZEPINE | E |

technique for predictive modeling, it is unsuitable for causal inference as it encodes the categorical variable using information from the outcome, which introduces a spurious, non-causal correlation. Our work utilizes a VAE (unsupervised) for non-linear dimensionality reduction, followed by PCA on the VAE's learned latent space to overcome this. This process extracts a few orthogonal components that explain over 90% of the variance, effectively capturing intricate relationships [17] and removing redundancy from the original categorical data. These resulting PCA-filtered embeddings serve as informative proxies for the categories in our causal effect estimation models. We hypothesize that these embeddings are also well-suited for modeling heterogeneous effects. Interpretability is maintained through a mapping from each embedding coordinate back to its original category level, with the VAE architecture detailed in Figure 1.

## C. Ensemble Methods and Bootstrapping in CSD

Previous studies that relied on bootstrapping have employed a fixed number of bootstrap replicates. For example, Wang and Peng [11] used 100 replicates in their DAGBag method. Li et al. [18] used 6 replicates in their C-EL method, and another study [19] employed 4 replicates. However, the justification for their chosen number of bootstraps is not clearly defined. Our ensemble method for causal inference adopts a three-stage process to ensure robustness and accuracy in the resulting causal network, with detailed procedures outlined in Algorithm 1 for Stage 1, Algorithm 2 for Stage 2, and Algorithm 3 for Stage 3. The initial step in this process involves applying several distinct CSD algorithms to generate a diverse set of candidate causal graphs representing **potential causal structures**, which then serve as inputs for our ensemble learning stages. For constraint-based CSD, we employ the Peter-Clark (PC) [20] algorithm, using several conditional independence tests appropriate for discrete data, including tests based on chi-squared, G-squared, Cressie-Read and Freeman-Tukey statistics. In the score-based approach, we implemented the Hill Climb algorithm [21] with various scoring functions suitable for discrete variables, namely Akaike information criterion (AIC), Bayesian information criterion (BIC), K2, and BDeu. Furthermore, we used the Fast Greedy Equivalence Search (FGES) algorithm [22], adapting it to use the BDeu score for our discrete data set. In Stage 1, the

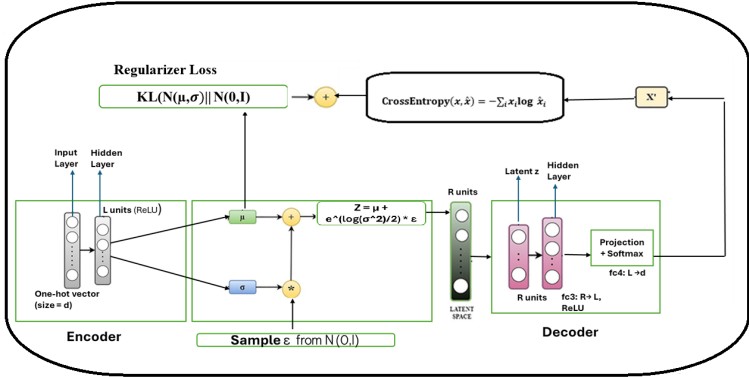

Fig. 1: Proposed VAE architecture for categorical encoding. Latent (R) and hidden (L) dimensions were tuned by minimizing validation loss.

optimal number of bootstrap replicates required for stable inference is adaptively determined. This involves iteratively generating bootstrap samples, applying multiple CSD (score-based) algorithms, and monitoring the convergence of edge confidence scores against a predefined threshold. Stage 2 utilizes the determined optimal number of bootstrap replicates. For each CSD algorithm (where feasible for bootstrapping), robust edge confidence scores are computed based on the frequency of edge occurrence. Conflicting edge directions are resolved using statistical tests (a binomial z-test, relying on the Central Limit Theorem [23] for the approximate normality of edge frequencies given a large number of replicates), and the resulting graph for each algorithm is ensured to be a DAG through cycle removal. However, due to the computational cost associated with extensive bootstrapping of the PC algorithm, we first learned its structure from the full data set. Edges within this learned PC structure were then reevaluated for statistical significance using an additional independence test function, and those found to be statistically significant were assigned an edge confidence of one for subsequent ensemble processing. Finally, stage 3 aggregates the outputs from the multiple algorithms. Composite confidence scores are computed for each potential edge, integrating factors such as its empirical frequency and average weight across algorithms from Stage 2, giving higher weight to score-based methods as per domain expert guidance on their closer alignment with existing knowledge. Any remaining conflicts in the edge direction are resolved, typically by favoring the edge with the higher composite score, to produce a final, high-confidence DAG representing the consensus causal structure. Although not direct inputs to the effect estimation models, these edge confidences critically inform the certainty of inferred edge directions and enable a targeted approach to cycle elimination.

For comparison with a domain knowledge graph developed with Mount Sinai Hospital clinicians, our learned ensemble causal graph (ECG) was evaluated using a composite score (see Table V). This score equally weighted (0.50 each) a structural component (average skeleton F1 [metric1] and orientation accuracy [metric2] [24]) and a path-based component

---

**Algorithm 1** Adaptive Bootstrapping (Stage 1)

**Require:** Dataset $D$; tolerance $c$=0.01; maximum iterations $M$=2k.
**Ensure:** Final edge confidence map *finalConf*; total bootstrap iterations $T_{final}$.
1: Initialize edge counts map *edgeCounts* $\leftarrow \emptyset$; *prevConf* $\leftarrow$ **nil**; total iterations $T \leftarrow$ 0.
2: **for** $iteration \leftarrow 1$ to $M$ **do**
3:    $D_{boot} \leftarrow$ BOOTSTRAPSAMPLE($D$)
4:    *model* $\leftarrow$ STRUCTURELEARNING($D_{boot}$)
5:    Update *edgeCounts* by incrementing counts for all edges found in *model*.
6:    $T \leftarrow T + 1$.
7:    Calculate current confidences: *currConf*[$e$] $\leftarrow$ *edgeCounts*[$e$]$/T$ for all $e \in$ *edgeCounts*.
8:    **if** *prevConf* $=$ **nil then**
9:       *maxChange* $\leftarrow \infty$
10:    **else**
11:       *maxChange* $\leftarrow \max_e \left| currConf[e] - prevConf[e] \right|$
12:    **end if**
13:    **if** *maxChange* $< c$ **then**
14:       **break**
15:    **end if**
16:    *prevConf* $\leftarrow$ *currConf*.
17: **end for**
18: *finalConf* $\leftarrow$ *currConf*; $T_{final} \leftarrow T$.
19: **return** (*finalConf*, $T_{final}$)

---

(average reachability Jaccard [metric3] [25] and shortest-path-length similarity [metric4]), prioritizing correct edge structure and direction while valuing overall connectivity.

*D. Causal Effect Estimation*

For causal effect estimation, we primarily relied on GLMs [26] applied within the backdoor adjustment framework. GLMs were selected due to their flexibility making them well-suited for our analysis as our outcome and most covariates were discrete. This approach allowed for the accommodation of response variables that do not follow a normal distribution. The ATE of an exposure A on an outcome Y, given an identified set of backdoor adjustment [27] [28] variables W, was calculated using the backdoor adjustment formula given in equation 2. The conditional expectation term was estimated using regression-based methods, specifically with GLMs serving as the regressor. Estimation of both $E[Y \mid A, W = w]$ and the weighting by $P(W = w)$ in this formula incorporated survey weights for national representativeness. Our backdoor adjustment relies on assuming: that all common causes (confounders) are measured, observed data indepen-

**Algorithm 2** Generalized Bootstrapped Causal Structure Discovery with Significance Testing (Stage 2)

**Require:** Dataset $D$, Protected Nodes $P$, Number of Bootstraps $N$, Confidence Threshold $T$=0.1, Significance Threshold $\alpha$=0.05
**Ensure:** DAG $G$ with Confidence-Weighted Edges
1: Construct blacklist $B$ based on Protected Nodes $P$ [age, gender and race]
2: Define CAUSALINFERENCE($D^*$, $B$) as the base structure learning function per bootstrap sample.
3: Initialize edge frequency map $\mathcal{E} \leftarrow \emptyset$
4: **for** $i = 1$ to $N$ **do**
5:     $D^* \leftarrow$ BOOTSTRAPSAMPLE($D$)
6:     $G_i \leftarrow$ CAUSALINFERENCE($D^*$, $B$)
7:     Update $\mathcal{E}$ by incrementing counts for edges in $G_i$
8: **end for**
9: Compute confidence $C(e) \leftarrow \frac{\mathcal{E}[e]}{N}$ for all edges $e \in \mathcal{E}$
10: Initialize candidate_edges $\leftarrow \emptyset$
11: **for all** each edge $e = (u, v)$ with $C(e) \geq T$ **do**
12:     **if** reverse edge $e' = (v, u)$ exists with $C(e') \geq T$ **then**
13:         Resolve conflict between $e$ and $e'$ using binomial z-test (level $\alpha$), add statistically dominant edge to candidate_edges with its confidence.
14:     **else**
15:         Add $(e, C(e))$ to candidate_edges
16:     **end if**
17: **end for**
18: Construct graph $G$ from candidate_edges with their confidences as weights.
19: **while** $G$ contains a cycle **do**
20:     Remove edge with minimum confidence from a detected cycle in $G$.
21: **end while**
22: **return** $G$

---

**Algorithm 3** Construction of Final Ensemble Graph (Stage 3)

**Require:** Input DAGs $G_1, \ldots, G_n$ with edge weights weight$_i(u, v)$;
    Algorithm weights $w_1, \ldots, w_n$ (PC variants: 0.5; score-based: 2); Confidence threshold $T$=0.1.
**Ensure:** Final ensemble DAG $G_f$ with composite edge confidences.
1: For each edge $(u, v)$ that appears in at least one input graph $G_k$ (where $k \in \{1, \ldots, n\}$), compute its composite confidence:
    $\text{conf}[u, v] \leftarrow \frac{\sum_{i:(u,v)\in G_i} w_i \cdot \text{weight}_i(u,v)}{\sum_{j=1}^{n} w_j}$
    (if the denominator $\sum_{j=1}^{n} w_j = 0$, then $\text{conf}[u, v] \leftarrow 0$; otherwise, if the numerator is 0, $\text{conf}[u, v]$ will also be 0).
2: Initialize candidate edge set $\mathcal{E} \leftarrow \{(u, v) \mid \text{conf}[u, v] \geq T \text{ and conf}[u, v] \text{ is defined}\}$.
3: Resolve conflicting directions in $\mathcal{E}$: for any pair $\{u, v\}$ where both $(u, v)$ and $(v, u)$ are in $\mathcal{E}$, remove the edge with the lower composite confidence.
4: Construct graph $G_f$ using edges in the refined $\mathcal{E}$ and their $\text{conf}[u, v]$ as weights.
5: **while** $G_f$ contains a cycle **do**
6:     Identify any directed cycle; remove its edge $e_{\min}$ with the smallest $\text{conf}[e_{\min}]$.
7: **end while**
8: **return** $G_f$

---

dencies accurately reflect the true causal graph structure, no deterministic treatment assignment by confounders, and our outcome regression model is correctly specified.

$$ACE = \sum_w E[Y \mid A, W = w]P(W = w) \tag{2}$$

*E. Experimental Setup*

Algorithms to discover causal structure were implemented using pgmpy [29] and bnlearn [30], visualized with networkx [31], and potential causal effects were estimated via DoWhy [32]. Optimal hyperparameters for VAE-based encoding of marital status (latent: 4, hidden: 32, lr: 0.0026) and race (latent: 3, hidden: 256, lr:0.0052) were determined using Optuna [33] by minimizing validation loss over 500 training epochs. For the PC algorithm, conditional independence tests used a 0.05 significance level. The specific parameters for Algorithms 1, 2, and 3 are detailed directly within their respective descriptions.

## III. RESULTS

*A. Mapping of Categorical Variables to VAE Embeddings, Number of Bootstraps and Generated Causal Graphs*

VAE-generated low-dimensional embeddings are shown in Table III. Here, we projected each category's VAE latent means down to one dimension via PCA and retained only the first principal component (PC1), which captures over 99% of the latent-space variance. The "VAE Embedding" column therefore reports each category's normalized PC1 score, which we then used directly in our causal-effect estimation models. Table IV shows how the maximum change in edge confidence scores diminishes and converges with an increasing number of bootstraps for the various algorithms and scoring functions utilized in our study. These plots highlight why a one-size-fits-all approach to selecting a fixed number of bootstrap samples is suboptimal and why our adaptive strategy is necessary.

TABLE III: VAE-generated embeddings

| Variable | Mappings_Before | VAE Embeddings |
|---|---|---|
| Race | Mexican American | 0.425461 |
| | Non-Hispanic Black | 1 |
| | Non-Hispanic White | 0 |
| | Other Hispanic | 0.434796 |
| | Other Race- Including Multi-Racial | 0.417866 |
| Marital Status | Married or Living with partner | 0 |
| | Widowed/Divorced/Separated | 1 |
| | Never Married | 0.426825 |

To compare the data-driven ECG shown in figure 2 with the domain expert DAG, we employed a single composite similarity score presented in table V. Our initial ECG was found to be missing some edges supported by domain knowledge, possibly due to potential algorithmic limitations from data discretization. Incorporating expert knowledge to refine a data-driven graph is a standard and necessary practice to ensure the final model is plausible and robust [34]. Consequently, these links, including MDD → CHD, MDD → DIABETES, CHD → SD, and DIABETES → CHD, were incorporated. Misoriented disease → income edges were corrected, ASM1→ASM2 links, reflecting medication overlaps were pruned. Edges from gender to outcomes summarize complex underlying biological mechanisms. This Expert-Augmented Causal Graph (EACG) is used for our causal effect estimation as it closely matches the expert graph while still being founded upon a structure largely learned directly from our observational data, providing an empirically driven basis for our inferences. Additionally, our data-driven graph included significant edges related to marital status, a distinction from the expert graph that focused more intensively on disease-specific links rather than demographic variables.

*B. Effect Estimates and Refutation Results*

Table VI presents estimated potential ATEs derived using backdoor adjustment sets identified from our EACG, accompanied by concise hypotheses. If the adjustment set includes a mediator on the causal pathway, that ATE represents a direct effect. Most of the effects were statistically significant.

TABLE IV: Summary of maximum change in edge-confidence ($\Delta_{conf}$) across bootstrap sample sizes.

| Algorithm | Start $\Delta_{conf}$ (bootstraps) | End $\Delta_{conf}$ (bootstraps) | Bootstrap range |
|---|---|---|---|
| Fast GES | 0.1000 (10) | 0.0100 (100) | 10 – 100 |
| Hill Climb (AIC) | 0.0950 (200) | 0.0099 (1000) | 200 – 1000 |
| Hill Climb (K2) | 0.0700 (200) | 0.0108 (1000) | 200 – 1000 |
| Hill Climb (BIC) | 0.0950 (200) | 0.0104 (1000) | 200 – 1000 |
| Hill Climb (BDeu) | 0.0550 (200) | 0.0109 (800) | 200 – 800 |

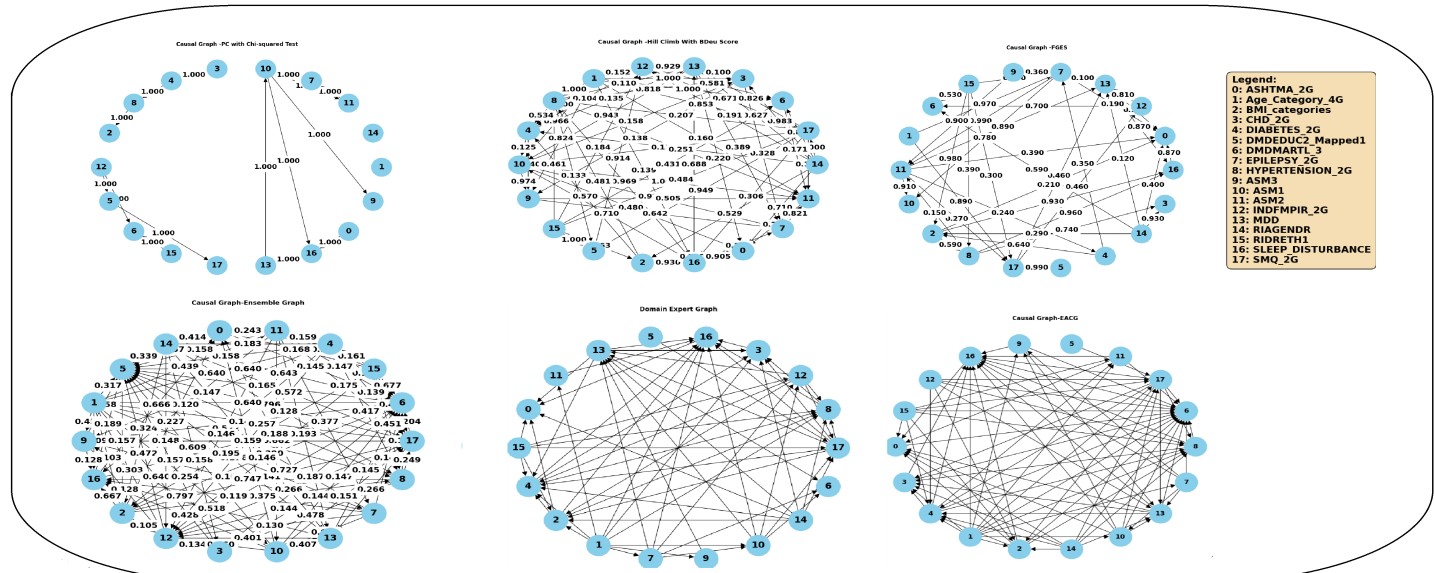

Fig. 2: Learned causal graphs: Representative outputs from PC, hill-climb (BDeu score), and FGES, alongside our initial weighted ensemble graph (bottom-left). In the ensemble graph, the directed links EPILEPSY_2G → SD and MDD → SD appear, indicating robust support for a potential causal effect of both epilepsy and depression on SD. Edges in the bottom-left graph with low confidence were retained because their presence is a direct result of aggregating evidence across multiple bootstrap samples. The final ensemble, expert-augmented structure, referred to as the EACG is in the bottom- right

TABLE V: Composite Score

| Graph | Metric1 | Metric2 | Metric3 | Metric4 | Score |
|---|---|---|---|---|---|
| ECG | 0.658 | 0.814 | 0.45 | 0.984 | 0.727 |
| EACG | 0.8368 | 1 | 0.8217 | 0.992 | **0.9127** |

However, the potential effect of mood-altering ASM3 on MDD, and the potential direct effect of epilepsy on MDD, were not statistically significant. Refutation tests as shown in the table VII supported the confidence in our significant findings. Dummy outcome refutation tests [35] consistently yielded insignificant effects (all p-values > 0.05), suggesting our results are not spurious. Simulating an unobserved confounder (UC) 'stress' [36] (effects: 0.05 on treatment, 0.3 on outcome) minimally altered the ATE for the epilepsy→SD link from 0.36 to 0.3289, indicating reasonable robustness to such unmeasured confounding. Methodologically, using VAE-derived embeddings to estimate epilepsy's impact on SD preserved key categorical information.

Health disparities between Non-Hispanic Black and Non-Hispanic White Americans in the contexts of both epilepsy and sleep are well-documented in the literature [37]. To investigate the causal effect of epilepsy on SD within these populations,

we employed and compared two distinct encoding strategies. Our findings show a high degree of consistency between the methods. The VAE-based model yielded an Average Treatment Effect (ATE) of 0.360, which closely aligns with the ATE of 0.326 from the traditional one-hot encoding model. This consistency extended to the subgroup level, with both methods identifying similar Conditional Average Treatment Effects (CATEs) for Non-Hispanic Whites (0.330 vs. 0.342) and Non-Hispanic Blacks (0.194 vs. 0.200).

Non-Hispanic White individuals were seen to have a larger estimated effect of epilepsy on SD 0.331 compared to Non-Hispanic Black individuals with an effect of 0.194 as seen from Table VIII. It could be potentially influenced by underlying differences in seizure frequency or reporting between these groups. These disparities lend additional support to our hypothesis that these embeddings can effectively model heterogeneous treatment effects, but they should be viewed as hypothesis-generating and not as definitive claims.

## IV. DISCUSSION & CONCLUSION

Our work established a data-driven yet expert-refined potential causal graph with clinical understanding, identifying

TABLE VI: Estimated Potential ATE and Corresponding Hypotheses. pp denotes percentage points.

| Test | Outcome | Treatment | Reference | Alternative | Estimate | 95% CI | p-value | Hypothesis |
|------|---------|-----------|-----------|-------------|----------|--------|---------|------------|
| T1 | SD | Epilepsy | No Epilepsy | Has Epilepsy | 0.362 | [0.284, 0.784] | **<0.001** | Managing epilepsy may reduce SD risk by 36.2 pp. |
| T2 | SD | MDD | No MDD | Has MDD | 0.167 | [0.099, 0.200] | **<0.001** | Treating MDD may reduce SD risk by 16.7 pp. |
| T3 | SD | MDD + BMI | Thin, no MDD | Overweight, MDD | 0.311 | [0.249, 0.342] | **<0.001** | Treating MDD in overweight individuals may reduce SD risk by 31.1 pp (vs. thin, no MDD). |
| | | | | Normal, MDD | 0.214 | [0.151, 0.240] | **<0.001** | Treating MDD in normal-weight individuals may reduce SD risk by 21.4 pp (vs. thin, no MDD). |
| T5 | SD | Age + Epilepsy | Young, no epilepsy | Middle-aged, epilepsy | 0.416 | [0.334, 0.822] | **<0.001** | Managing epilepsy in middle-aged adults may reduce SD risk by 41.6 pp (vs. young, no epilepsy). |
| | | | | Older, epilepsy | 0.462 | [0.434, 0.902] | **<0.001** | Managing epilepsy in older adults may reduce SD risk by 46.2 pp (vs. young, no epilepsy). |
| T6 | SD | ASM2 | No ASM2 | Has ASM2 | 0.206 | [0.178, 0.321] | **<0.001** | Avoiding sleep-altering ASMs (ASM2) may reduce SD risk by 20.6 pp. |
| T7 | MDD | ASM3 | No ASM3 | Has ASM3 | 0.1611 | [0.037, 0.289] | 0.082 | Mood-altering ASMs (ASM3) show no significant effect on MDD risk. |
| T8 | MDD | Epilepsy | No Epilepsy | Has Epilepsy | 0.016 | [-0.196, 0.109] | 0.335 | Epilepsy shows no significant direct effect on MDD risk. |
| T9 | SD | Hypertension | No hypertension | Has hypertension | 0.104 | [0.086, 0.154] | **<0.001** | Treating hypertension may reduce SD risk by 10.4 pp. |

TABLE VII: Refutation Results (Estimated effect (EE) and New Effect (NE) with Dummy Outcome(DO) Refuter.

| Test | Refuter | EE | NE | p-value |
|------|---------|-----|-----|---------|
| T1 | DO | 0 | 0.156 | 0.09 |
| T1 | UC | 0.362 | 0.328 | – |
| T2 | DO | 0 | 0.01 | 0.72 |
| T3 (Overweight) | DO | 0 | -0.043 | 0.91 |
| T3 (Normal weight) | DO | 0 | -0.029 | 0.87 |
| T5-Middle age | DO | 0 | 0.13 | 0.17999 |
| T5-Old age | DO | 0 | 0.10 | 0.27 |
| T6 | DO | 0 | 0.019 | 0.59 |
| T9 | DO | 0 | 0.006 | 0.94 |

TABLE VIII: Sociodemographic Heterogeneous effect of epilepsy on SD by VAE-embedding bins, where bins and effects were identified during Conditional Average Treatment Effect (CATE) estimation

| Treatment | Outcome | Bins | Effect |
|-----------|---------|------|--------|
| Epilepsy | Sleep_Disturbance | [-0.001, 0.418] | 0.3313 |
| | | [0.418, 0.425] | 0.237 |
| | | [0.425, 1.0] | 0.194 |

epilepsy and MDD as potential direct causes of SD. Further examination of causal effects revealed that certain ASMs known to alter sleep architecture (e.g., benzodiazepines) were also associated with increased SD, in line with clinical reports that they deepen Stage 2 sleep at the expense of restorative slow-wave and REM phases. Mood-altering ASMs had no statistically significant causal impact on MDD in our sample, with both medication-related findings aligning with expert observations. The potential causal effect estimations also uncovered notable interactions, pinpointing heightened SD in older adults with epilepsy (consistent with epilepsy's age-related incidence peaks) and in individuals suffering from both obesity and MDD (reflecting adiposity's known adverse impact on sleep via metabolic and airway mechanisms). We also hypothesize that the discretization of epilepsy and MDD variables could be a contributing factor to the observed lack of statistical significance for these particular effects, and that continuous measures (e.g., full scores from questionnaire) might better capture these relationships. While these specific findings are hypothesis-generating from a plausible causal structure, they strongly suggest avenues for targeted interventional trials. Our findings can help inform clinical practice by enabling a shift from reactive to proactive care. For instance, clinicians can use our quantified potential causal links to choose an antiseizure drug with a lower risk of causing sleep disturbance. Furthermore, these results provide a framework for prompting timely comorbidity screening and creating personalized follow-up plans.

## V. LIMITATIONS AND FUTURE WORK

While this study offers initial insights, its limitations primarily stem from the use of cross-sectional NHANES data. It necessitates future longitudinal or interventional studies to firmly establish causal directionality, address potential reverse causation, and validate findings that rely on standard causal discovery assumptions (under section II-D), despite refutation tests for robustness against potential bias from omitted variables, such as seizure severity or family history. Further limitations include the generalized treatment of ASMs and the discretization of continuous variables like BMI and age. Additionally, our custom MDD index deviates slightly from the standard PHQ-9, limiting its direct comparability. Finally, the reliance on self-report questionnaires for assessing both MDD and SD may introduce measurement bias. Consequently, future research should prioritize longitudinal designs to delineate better temporal relationships and predictive influences.

We also advocate for incorporating more objective measures (e.g., actigraphy, polysomnography for sleep, detailed clinical assessments for epilepsy and depression) for enhanced validation. Future work will also explore applying specialized survey statistics methods to ensure the most robust variance estimation. We also plan on employing a causally-informed Multiple Imputation by Chained Equations (MICE) approach to replace the missing records.

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
