# OpenReview forum: "Understanding the Impact of Epilepsy and Depression on Sleep Disorder: Beyond Associations"
_IEEE.org/EMBS/BHI/2025/Conference — BHI 2025_

### Official Review · Reviewer_9w4b · 2025-06-29
**This paper proposes a novel framework for uncovering potential causal relationships among epilepsy, depression (MDD), and sleep disturbance (SD) using cross-sectional NHANES data. It would benefit from additional and/or longitudinal dataset to confirm the effectiveness and would present a strong foundation for future interventional studies.**

**Confidence:** 3
**Clarity Of Writing:** good
**Clinical Significance:** good
**Methodological Novelty:** great
**Overall Rating:** 6

**Experiments And Results:**

good

**Questions For The Authors:**

Q1: What's the likelihood of expert DAG conflicts the ensemble DAG? How do you resolve it?

Q2: Given your Algorithm 1, how robust is the ensemble DAG to minor perturbations in the data?

Q3: How do you confirm if your model is not over-engineered or overfitting ?

**Strengths:**

1. Innovative causal discovery pipeline: The ensemble CSD with adaptive bootstrapping and VAE-PCA encoding is technically sophisticated and methodologically robust for observational data.

2. Data-driven yet expert-informed: The authors effectively bridge empirical DAG inference with clinical domain knowledge by integrating expert corrections in the final causal graph (EACG).

3. Strong robustness checks: Use of refutation techniques (e.g., dummy outcome and simulated confounding) enhances confidence in the inferred effects.

4. Heterogeneous treatment effects: The authors make a strong case for the use of VAE embeddings to uncover group-specific variations in causal relationships.

5. Clear motivation and clinical relevance: Addressing the epilepsy-MDD-SD triad has direct implications for patient care and medication management.

**Summary Of The Paper:**

This paper explores the complex interplay among epilepsy, major depressive disorder (MDD), sleep disturbance (SD), and the role of antiseizure medications (ASMs), using cross-sectional data from NHANES (2015–2020). Recognizing that conventional statistical methods are limited in teasing apart associations versus causation in complex health data, the authors propose a multi-stage ensemble framework for causal structure discovery (CSD).

The core methodology includes:
1. Encoding high-cardinality categorical variables via a VAE-PCA pipeline.
2. Constructing a consensus Directed Acyclic Graph (DAG) using an ensemble of CSD methods with adaptive bootstrapping.
3. Estimating backdoor-adjusted Average Treatment Effects (ATEs) using Generalized Linear Models (GLMs).
4. Validating robustness via refutation tests, including dummy outcome and simulated confounding.

The author conclude that:
1) Epilepsy and MDD are potential direct causes of SD.
2) Sleep-altering ASMs also contribute to SD.
3) Mood-altering ASMs and epilepsy were not found to significantly affect MDD directly.
4) The refined DAG aligns well with expert-validated clinical graphs and reveals testable causal hypotheses, laying groundwork for future longitudinal studies.

**Weaknesses:**

1. Some figures and tables need to be reprinted. Font sizes are too small and unclear. Also please double check some minor typos and missing punctuations (e.g., Page 2 "... dataset such as the NHANES.", "The Global, regional, ...", Page 4, Algorithm 1 "(Stage 1", etc.)

2. some concerns may arise, such as would there be dependence on the dataset without cross validation? Since the dataset lacks temporal information, so the possibility of reverse causality remains. The method is very interesting, but the outcome may not be very confident enough due to the dataset.

---

### Official Review · Reviewer_gtek · 2025-07-04
**Understanding the Impact of Epilepsy and Depression on Sleep Disorder: Beyond Associations**

**Confidence:** 4
**Clarity Of Writing:** fair
**Clinical Significance:** good
**Methodological Novelty:** great
**Overall Rating:** 6
**Final Rating:** 6

**Experiments And Results:**

good

**Questions For The Authors:**

1.How do you see these causal findings being used in day-to-day clinical decision-making? Could you outline any specific interventions or policy recommendations that flow from your results?

2.What concrete benefit do the VAE-PCA embeddings give over ordinary one-hot encoding?

3. Table IV presents “Metric 1-Metric 4” but their meanings (skeleton F1, orientation accuracy, reachability Jaccard, shortest-path-length similarity) are only defined in the Methods section. Would you consider adding a cross-reference such as “(see Table IV)” next to metric definition in the Methods?

**Strengths:**

1.	Novel adaptive-bootstrap ensemble for causal structure discovery; parameters are documented for reproducibility.
2.	VAE-PCA embeddings replace one-hot encoding for unordered categorical covariates, reducing dimensionality.
3.	This study explores Conditional Average Treatment Effects (CATE) using VAE embeddings, revealing differential impacts of epilepsy
         on sleep disturbance across racial subgroups. This demonstrates attention to heterogeneous treatment effect.
4.	Highlights under-explored links between epilepsy, sleep disorders, depression, and drug effects.

**Summary Of The Paper:**

This study builds an expert-augmented causal graph—created by an adaptive ensemble of structure-discovery algorithms—to examine how epilepsy, major-depressive disorder (MDD), and sleep-disturbance (SD) influence one another, while accounting for sleep-altering antiseizure medications (ASMs). Average‐treatment-effect estimates (via survey-weighted GLMs) suggest sizeable causal links: epilepsy → SD (0.362), MDD → SD (0.167), and ASM class → SD (~0.206).

**Weaknesses:**

1.Generalization: The results, while promising, are based on specific datasets. There is a need for further validation across different populations and clinical settings to ensure generalization.

2. Several figures and tables are difficult to read, which detracts from the overall clarity and accessibility of the paper.

---

### Official Review · Reviewer_AQzv · 2025-07-15
**Review 21**

**Confidence:** 4
**Clarity Of Writing:** great
**Clinical Significance:** good
**Methodological Novelty:** good
**Overall Rating:** 5

**Experiments And Results:**

good

**Questions For The Authors:**

Have you tried benchmark VAE-PCA against simpler encodings such as target or entity embeddings, and if so, how did downstream causal estimates shift?

**Strengths:**

The study has strengths in methodological development and clinical grounding. Using VAE-PCA to interpret categorical variables is a clever way to handle dimensionality without losing signal. The ensemble-plus-bootstrap approach yield a stable causal graph than any single algorithm. Bringing in domain experts at the end keeps the model biologically plausible. By working with a representative cohort, the authors ensure the findings have public-health relevance.

**Summary Of The Paper:**

The authors analyze 2015-2020 NHANES data to explore the causal triangle linking epilepsy, sleep disturbance, and major-depressive disorder, with a special look at how different classes of anti-seizure medications (ASMs) might impact the balance. They compress high-cardinality categorical variables with a VAE-PCA pipeline, build an ensemble of causal-discovery algorithms (PC, score-based hill-climb, and FGES) that they stabilize through adaptive bootstrapping, then let a neurology–sleep-medicine panel tweak the resulting graph so it passes clinical common-sense. Average treatment effects are estimated with survey-weighted generalized linear models that satisfy the back-door criterion, and refutation tests probes how robust the inferences are to hidden bias. The headline results: epilepsy itself and ASMs known to disrupt sleep architecture both appear to be direct causes of sleep disturbance; epilepsy does not seem to cause depression once covariates are controlled; and mood-altering ASMs show no independent link to depression within this dataset. A public DAG, code, and an unusually candid limitations section round out the contribution.

**Weaknesses:**

Sleep disturbance and depression are self-reported, which invites recall and reporting bias, and no polysomnography or PHQ-9 raw scores are used to validate or quantify severity.

List-wise deletion of incomplete records may skew the sample, while collapsing continuous variables (age, BMI) and classifying ASMs into only three binary flags flatten potentially important dose-response effects and drug interactions.

Manual edits to the ensemble DAG, plus subjective weighting of the underlying algorithms, could inject bias.

Finally, because seizure frequency and duration are missing, unmeasured disease severity might still confound several of the reported effects.

---

### Official Review · Reviewer_rkzB · 2025-07-17
**paper revision**

**Confidence:** 4
**Clarity Of Writing:** good
**Clinical Significance:** good
**Methodological Novelty:** fair
**Overall Rating:** 6

**Experiments And Results:**

good

**Questions For The Authors:**

The paper's idea is on effect estimates derived from the EACG, a graph that was manually edited to conform to existing knowledge. How can any finding from this graph be presented as feature?

what is the reason to use that complex and non-interpretable VAE-PCA encoding? and why it is superior to standard methods like one-hot encoding with interaction terms?

**Strengths:**

The paper have clinically good contents. The motivation to use these relationships is strong and highly relevant.
The effort to move beyond correlational analysis and apply a causal inference framework to complex observational data is interesting.
The inclusion of refutation tests, such as the dummy outcome refuter and sensitivity analysis for an unobserved confounder, is a notable strength that shows methodological interests.

**Summary Of The Paper:**

Based on my review, this paper have the complex, potentially causal relationships between epilepsy, MDD, and SD using multiple data sources.
Paper employ a multi-stage methodology to move beyond simple association statistics. they process the data, using a Variational Autoencoder and PCA pipeline to create low-dimensional categorical variables like race and marital status. they use ensemble of causal structure discovery algorithms with an adaptive bootstrapping procedure to learn a data-driven Directed Acyclic Graph

**Weaknesses:**

The single greatest weakness is the process of creating and then relying on Expert-Augmented Causal Graph.
This methodology is a bit ambiguous and circular. It uses the data to generate graph structure, but when the structure is not as expected, it is manually changed. does not represent a discovery from the data.
The paper use VAE-PCA pipeline to encode categorical variables, as a novel method. However, this only adds complexity with little benefit.

---

### Official Review · Reviewer_rLG7 · 2025-07-21
**Understanding the Impact of Epilepsy and Depression on Sleep Disorder: Beyond Associations**

**Confidence:** 3
**Clarity Of Writing:** excellent
**Clinical Significance:** excellent
**Methodological Novelty:** excellent
**Overall Rating:** 8

**Experiments And Results:**

good

**Questions For The Authors:**

No question for the authors here.

**Strengths:**

The paper is well written and the methodology sounds very attractive ! The authors have tackled one of the major problem of the literature by building a strong methodology.
I liked the ECG generation for the purpose of the work.

**Summary Of The Paper:**

This study uses NHANES 2015-2020 (pre-pandemic) data to formulate robust causal hypotheses on the interaction between epilepsy, SD, MDD and ASMs. Categorical variables were processed with a VAE-PCA pipeline. A directed acyclic graph (DAG, ECG) was constructed via a set of causal structure discovery algorithms, then adjusted using medical expertise.

**Weaknesses:**

It is regrettable that all the tables or figures proposed are too small ! The authors should reduce a bit their first introductive part to leave room to readable figures and tables values.
The abstract is little bit difficult to read such the methodology is complex. Maybe, shorten the ideas could improve this.
Be careful not writing a lot of stuff in the perspective work section. The authors should select one or maximum two ideas by selecting the ones which could be feasible in reasonable time.